# Influence of follow-up, screening age, interval, and compliance on overdiagnosis of ductal carcinoma in situ (DCIS): A modelling study

Keris Poelhekken[1,2]*, Marcel J.W. Greuter[2], Monique D. Dorrius[2], Geertruida H. de Bock[1]

1 Department of Epidemiology, University of Groningen, University Medical Center Groningen, Groningen, The Netherlands, 2 Department of Radiology, University of Groningen, University Medical Center Groningen, Groningen, The Netherlands

* k.poelhekken@umcg.nl

## Abstract

Overdiagnosis estimates of ductal carcinoma in situ (DCIS) vary from 20–91%, which complicates screening communication and optimization. The aim was to quantify the influence of follow-up time and screening setting on overdiagnosis of DCIS. The fully validated micro-simulation Markov model for DCIS (SimDCIS) was used to estimate DCIS overdiagnosis in different screening settings with varying follow-up. DCIS overdiagnosis was defined as the number of diagnosed DCIS (screen-, clinically detected, or progressed to invasive breast cancer) in screening that would not have been diagnosed without screening. Outcomes were presented as overdiagnosed proportion and rate. The base cohort was screened biennially from age 50–74 with 76% compliance and 25 years (y) follow-up and compared to the non-screened cohort. Follow-up was varied from 2-25y, screening start 40-74y, screening interval 1-5y and compliance 50–100%. DCIS overdiagnosis was estimated at 20% of all diagnosed DCIS and 38.1 overdiagnosed DCIS per 100,000 women screened biennially from age 50–74 at 76% compliance and 25y follow-up. The proportion of overdiagnosed DCIS increased with shorter follow-up (27% at 2y to 20% at 25y), older screening start age (1% at 40y to 15% at 74y), decreased screening interval (23% at 1y to 12% at 5y), and increased compliance (16% at half to 20% at full participation). In conclusion, reliable DCIS overdiagnosis estimates require attention to screening setting and ≥20 years follow-up. Older women (74y) showed up to seven times more overdiagnosis at initial screening than younger women (50y). Improved estimates can provide guidance in screening communication and optimization.

## Introduction

Worldwide, breast cancer is the most diagnosed cancer in women and often detected at an early stage [1,2]. Whereas ductal carcinoma in situ (DCIS) was hardly diagnosed

**Data availability statement:** The SimDCIS model v02 was used and is publicly available in GitHub at https://github.com/kp-gith/SimDCIS, in programming language C++.

**Funding:** The author(s) received no specific funding for this work.

**Competing interests:** The authors have declared that no competing interests exist.

before introduction of screening, it nowadays accounts for approximately 25% of all detected breast cancer cases in countries with a screening program [3–5]. DCIS is considered as stage zero breast cancer and a precursor of invasive breast cancer (IBC). It can be divided into DCIS grade 1, 2, and 3, where a higher grade is more likely to progress to IBC [3,5,6]. The increased detection of DCIS has raised concerns for overdiagnosis and overtreatment related to breast cancer screening, which highlights the need for optimization, including early-detection and overdiagnosis of DCIS [3,4]. Overdiagnosis refers to the detection of breast cancer in screening that would not have been diagnosed without screening [5,7]. Overtreatment refers to treatment without reduction of breast cancer mortality risk. DCIS overdiagnosis often results in overtreatment, as generally DCIS is still treated upon detection [3,5]. Overdiagnosis is considered the main harm of screening and estimates for DCIS range from 20–91% [2,3]. This large uncertainty in estimates complicates optimization of screening, as the actual impact of DCIS overdiagnosis as major harm cannot be accurately stated. This large range is partly the result of the unknown natural history of DCIS [5,8] and of the large variation in overdiagnosis definitions [9,10]. Overdiagnosis estimates also highly depend on the determinants considered within the estimate. An increase in DCIS grade is associated with increased overdiagnosis estimates [10,11].

The effect of multiple important determinants on DCIS overdiagnosis estimates has not yet been established. Determinants associated with IBC-related overdiagnosis are follow-up time and screening setting (Table 1). One study estimated the effect of age on DCIS overdiagnosis, with lower overdiagnosis for women with DCIS grade 3 aged 50–60 (21–29%) compared to women aged 60–75 (50–66%) [11]. Furthermore, IBC-related overdiagnosis decreases with increased follow-up time due to the lead-time effect [12,13]. For DCIS, length of follow-up needed to account for lead time has not been established yet. In addition, screening interval affects total breast cancer overdiagnosis (DCIS and IBC) and is expected to affect DCIS overdiagnosis also, as screening interval affects the detection probability of DCIS [14,15]. Also, the effect of compliance on the extent of overdiagnosis should be established, as participation rates declined over the past 10 years in multiple countries (e.g., from 82% in 2008 to 72% in 2021 in the Netherlands), which affects the balance between benefits and harms of screening [16]. Lower compliance (80% versus 100%) has been shown to lower overdiagnosis estimates of total breast cancer by approximately 5% [14].

Improved knowledge on the influence of each determinant on DCIS overdiagnosis estimates can help to understand the previously reported wide range of 20–91% overdiagnosed DCIS and improve accuracy of these estimates. These more accurate overdiagnosis estimates can be used to optimize breast cancer screening and treatment policy by making it feasible to weigh the benefit of early detection and the harm of DCIS overdiagnosis. Simulation models can, if data is available, account for important determinants in the estimation of overdiagnosis [17]. SimDCIS is a previously constructed, fully validated, robust simulation model, which can make accurate age- and grade-dependent estimations in screening setting [15]. Therefore, the aim of this study was to quantify the influence of follow-up time, screening age, interval, and compliance on overdiagnosis using the fully validated SimDCIS model.

**Table 1. Established determinants of breast cancer overdiagnosis.**

| Determinant | Influence on overdiagnosis estimate | | Breast cancer | Reference |
|---|---|---|---|---|
| increase | Direction | Approximate magnitude | | |
| Follow-up time | decrease | 3% screened from 2y to 15y | IBC | [12] |
| Screening start age | increase | 263/100,000 screened from 50y to 68y | IBC | [12] |
| Screening age | increase | 30% from 50-60y to 60-75y | DCIS grade 3 | [11] |
| Screening interval | decrease | 9% from annual to triennial | Total | [14] |
| Screening compliance | increase | 5% from 80% to 100% | Total | [14] |
| DCIS grade | decrease | 6% from grade 1–3<br>15% from grade 1–3 | DCIS<br>DCIS | [10]<br>[11] |

Overview of established determinants of breast cancer overdiagnosis for invasive breast cancer (IBC), ductal carcinoma in situ (DCIS), and total breast cancer (IBC+DCIS), including the direction and magnitude of their influence on overdiagnosis estimates.

## Materials and methods

### The SimDCIS model

The SimDCIS model is a micro-simulation Markov model, in which women are simulated on an individual level and transition between states during their lifetime [15]. Within a simulation, a virtual cohort of 100,000 women that mimics a specific population is created and followed from birth until one of the following end states: death, screen-detected DCIS, clinically detected DCIS, or presence of IBC. Input parameters of SimDCIS include four transition probabilities (healthy to death, healthy to DCIS, DCIS to IBC, regression of DCIS to healthy), four screening parameters (sensitivity, age, interval, and compliance), and a probability for clinical detection of DCIS. SimDCIS can be applied to a specific population by adjusting death probability, clinical detection, sensitivity, screening age, interval, and compliance. The model was previously applied to and successfully validated in Dutch and United Kingdom screening setting and full robustness and uncertainty analyses for overdiagnosis were performed [10]. All input parameters were independently derived from literature [15]. The death probability was assumed equal for women in the healthy and DCIS states [8,17]. DCIS onset probability was age- and grade-dependent and based on data from the Netherlands Cancer Registry of 2015–2022 (excluding 2020 due to the COVID-19 pandemic) [18]. The probability of DCIS progression to IBC was derived from age-dependent data from the United States National Cancer Institute's Surveillance, Epidemiology, and End Results program of 1992–2014 [19]. Direct progression from healthy to IBC was not included in SimDCIS, as the main goal of the model was to estimate DCIS overdiagnosis which is not affected by IBC without DCIS as a precursor. The exclusion of direct progression to IBC, which was estimated at 18% of all IBC [20], resulted in a larger healthy population compared to real-world. To estimate the effect of this assumption on our overdiagnosis estimate, an additional sensitivity analysis was performed with 82% of the population. Regression of DCIS to the healthy state was included at 5% per year [17,21], but regression of IBC to DCIS was not included as this is not supported by tumour biology research [17,22]. DCIS could be detected by mammography, with a sensitivity of 86% per screening [23], or clinical detection, with a probability of 5% per year [11].

### Base scenario and definitions

The SimDCIS model was applied to the Dutch breast cancer screening setting as base scenario, as this is a well-implemented screening program of which high quality data is available. In Dutch screening setting, women aged 50–74 years are invited to a biennial examination with mammography, with a compliance of 76% in 2019 [24]. Age-dependent death probability in the Netherlands was derived from data of the Central Bureau of Statistics [25].

The main outcomes were the proportion and rate of overdiagnosed DCIS (S1 File). The definition of DCIS overdiagnosis varies in literature. Previous recommendations highlight that estimates of DCIS overdiagnosis should account

for progression to IBC, include clinical detection, and should be made from population or individual perspective [26]. In our previous publication, a definition for DCIS overdiagnosis was recommended [10], which was also used in this study. Overdiagnosed DCIS was defined as the excess of diagnosed DCIS in a screened cohort compared to an unscreened cohort. Diagnosed DCIS included screen-detected DCIS, clinically detected DCIS, and DCIS that progressed to IBC. The screened and unscreened cohorts contained one-to-one identical women, and the simulations with SimDCIS were fully identical, except for the presence of screening. Because women in the screened and non-screened cohorts were identical, differences in DCIS detection and survival per woman could be determined, and overdiagnosis per woman and cohort could be estimated. The proportion of overdiagnosed DCIS quantifies the risk of a DCIS diagnosis being overdiagnosed and was calculated as the number of overdiagnosed DCIS divided by all diagnosed DCIS in the screened cohort. The overdiagnosis rate quantifies the risk of overdiagnosis for a women going to screening and was calculated as the number of overdiagnosed DCIS per 100,000 screened women. All outcomes were calculated from a population perspective for women from 50–100 years, and averaged over 10 cohorts of 100,000 women.

### Determinants

The influence of follow-up time, screening start age, screening interval, and compliance were quantified on overdiagnosis estimates, overall and stratified by DCIS grade. The follow-up time was varied from 2–25 years after screening. The screening start age was varied from 50–74 years every 2 years, with a single screening round simulated for each screening age. In addition, screening start age was stratified by DCIS grade to establish whether the effect of age was grade dependent, and evaluated from 40–49 to explore the effect of an earlier screen start age in line with guidelines worldwide [27,28]. Screening interval was varied from one to five years in steps of one year. Compliance was varied from 50%−100% in steps of 10%. Overdiagnosis was quantified for all combinations of screening interval and compliance to evaluate overdiagnosis in different screening settings.

### Ethics statement

This study used publicly available fully anonymized retrospective aggregated data; no individual data was used. Our local institutional review board (IRB) confirmed this study was exempt from ethical compliance and waived the requirement for informed consent (IRB number M24.343483, Medical Ethics Review Board Groningen, The Netherlands).

## Results

### Follow-up time

Overall, for women aged 50–74 screened biennially, overdiagnosis was estimated at 27.0% of all detected tumours in a screened population and 48.6 per 100,000 screened women at 2 years follow-up, and decreased to 19.6% and 38.1 at 25 years follow-up (Table 2 and S1 Table). The decrease in overdiagnosis with increased follow-up time was 8.0%, 8.1%, and 6.6% for DCIS grade 1, 2, and 3, respectively (Table 2).

### Screening start age

DCIS overdiagnosis increased with 13% from age 50–74 years at 25 years follow-up (Fig 1). DCIS overdiagnosis was estimated at 37.8 per 100,000 screened and 2.4% of all DCIS in a screened population at age 50, rising to 112.1 and 15.4% for age 74 (Fig 1 and S1 Fig). A similar effect for follow-up time was found, with a decrease in overdiagnosis estimate with longer follow-up for all screening start ages between 50 and 74 years. The influence of age on the proportion of overdiagnosed DCIS was larger for a longer follow-up. With 2 years follow-up, the estimated proportion overdiagnosed DCIS showed a difference of 2.5% between screening start age 50 and 74 compared to 13% at 25 years follow-up. Similar effects were observed for age 40–49 (S2 Table) with the lowest overdiagnosed proportion of 0.6% at 25 years

**Table 2. Influence of follow-up time on DCIS overdiagnosed proportion.**

| *Proportion overdiagnosed* | DCIS[a] Grade | | | |
|---|---|---|---|---|
| **Follow-up time** | **All** | **1** | **2** | **3** |
| **2 years** | 27.0% | 31.6% | 28.1% | 24.1% |
| **3 years** | 25.3% | 29.9% | 26.1% | 22.6% |
| **4 years** | 23.9% | 28.8% | 24.7% | 21.1% |
| **5 years** | 22.9% | 27.8% | 23.5% | 20.2% |
| **10 years** | 20.6% | 25.1% | 21.0% | 18.3% |
| **15 years** | 19.8% | 23.9% | 20.3% | 17.7% |
| **20 years** | 19.6% | 23.7% | 20.0% | 17.6% |
| **25 years** | 19.6% | 23.6% | 20.0% | 17.5% |

Overall overdiagnosed proportion DCIS (per diagnosed in screening) and stratified by grade for a follow-up of 2–25 years in Dutch screening setting (biennial, age 50–74, 76% compliance). [a]DCIS = ductal carcinoma in situ.

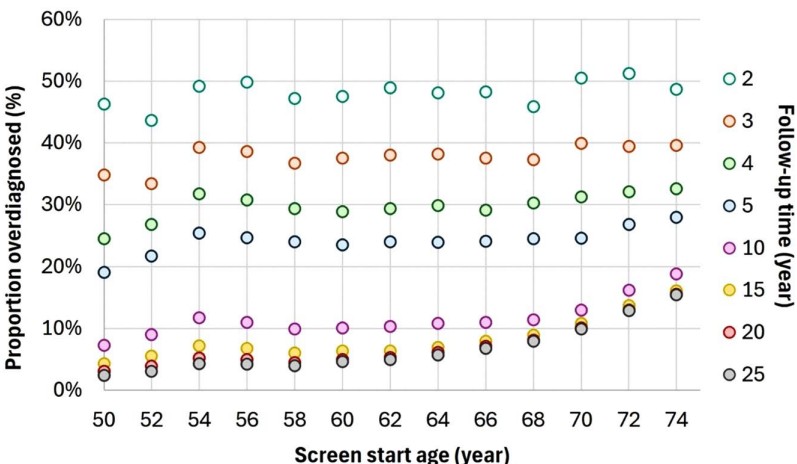

**Fig 1. Influence of screening start age on DCIS overdiagnosed proportion with varying follow-up time.** Proportion overdiagnosed DCIS after a single screen at age 50-74 years for a follow-up of 2 to 25 years in Dutch screening setting (biennial mammography, 76% compliance).

follow-up, and stratified by DCIS grade, with a consistently highest overdiagnosis for grade 1 and lowest for grade 3 (S3 and S4 Tables).

## Screening interval and compliance

The proportion overdiagnosed DCIS decreased with 10.7% for an increased screen interval of 1–5 years for women aged 50–74 at 76% compliance with 25 years follow-up (Table 3). Contrarily, DCIS overdiagnosis rate increased to 29.8 per 100,000 screened for an increase in interval of 1–5 years (S5 Table). In biennial setting, an increase in screening compliance of 50% to 100% resulted in a decreased overdiagnosis rate from 45.3 to 33.5 and an increased proportion overdiagnosed of 16.0% to 22.0%, respectively. The influence of screen interval on overdiagnosis was larger for a higher compliance. For an increase in screen interval of 1–5 years, DCIS overdiagnosis rate per 100,000 screened women increased with 26.7 at 50% compliance and with 31.4 at 100% compliance, whereas the proportion overdiagnosed DCIS decreased with 11.4% at 50% compliance and 9.7% at 100% compliance. The influence of the studied determinants on DCIS overdiagnosis is summarized in Table 4.

**Table 3. Influence of screening interval and compliance on DCIS overdiagnosis.**

| Screen interval (years)[a] | Compliance (%) | | | | | | |
|---|---|---|---|---|---|---|---|
| | 50 | 60 | 70 | 80 | 90 | 100 | 76 |
| *Proportion overdiagnosed (per detected DCIS in screened population)* | | | | | | | |
| 1 | 20.0% | 21.2% | 22.1% | 22.9% | 23.5% | 24.1% | 22.5% |
| 2 | 16.0% | 17.6% | 18.9% | 20.1% | 21.1% | 22.0% | 19.6% |
| 3 | 13.3% | 15.0% | 16.5% | 17.7% | 18.8% | 19.8% | 17.2% |
| 4 | 11.8% | 13.4% | 14.9% | 16.2% | 17.4% | 18.5% | 15.6% |
| 5 | 8.6% | 9.9% | 11.1% | 12.2% | 13.4% | 14.4% | 11.8% |

Overdiagnosed proportion DCIS for women aged 50–74 with screening interval of 1–5 years for varying compliance of 50–100% and 76% of the Dutch screening setting.

**Table 4. Summary influence determinants of DCIS overdiagnosis.**

| Influence on | Proportion overdiagnosed | | Overdiagnosis rate/ 100,000 screened | | |
|---|---|---|---|---|---|
| Determinant for increase in | Direction | Approximate magnitude | Direction | Approximate magnitude | |
| Follow-up time | decrease | 7% | decrease | 10 | from 2y to 25y |
| Screening start age | increase | 13% | increase | 74 | from 50y to 74y |
| Screening interval | decrease | 11% | increase | 30 | from 1y to 5y |
| Screening compliance | increase | 6% | decrease | 12 | from 50% to 100% |

Determinants of DCIS overdiagnosis estimated in the current study, including the direction and magnitude of their influence on overdiagnosis estimates.

## Discussion

For a diagnosed DCIS, the probability of the DCIS case to be overdiagnosed was estimated at 20%, and for 100,000 screened women 38.1 cases of DCIS were overdiagnosed with 25 years follow-up after the last screening round in Dutch screening setting (biennial, age 50–74, 76% compliance). A longer follow-up decreased overdiagnosis estimates overall and by grade with approximately 7% (Table 4), and estimates stabilized from 20 years of follow-up. The proportion overdiagnosed DCIS at screen start age 74 was over seven times higher compared to age 50 with 25 years follow-up (2.4% to 15.4%), whereas the age effect diminished for shorter follow-up. An increased screen interval from 1 to 5 years resulted in an 11% decrease in the proportion overdiagnosed DCIS, but an increase in overdiagnosis rate of 29.8 per 100,000 screened women. An increased compliance (50% to 100%) increased the proportion overdiagnosed by 6% and decreased overdiagnosis rate by 11.8. The influence of screen interval on overdiagnosis increased with a higher compliance.

Follow-up time causes an overestimation of DCIS overdiagnosis estimates, with overestimation up to 7% and 10/100,000 screened women with insufficient follow-up time. Sufficient follow-up time was established at 20 years, independent of DCIS grade. Overestimation of overdiagnosis at short follow-up can be explained by the lead-time effect [12,29], but the magnitude of overestimation and minimum required follow-up time to avoid overestimation had not yet been established for DCIS overdiagnosis. Previous studies have shown a similar effect for IBC-related overdiagnosis, with a decrease in overdiagnosis estimates for longer follow-up [12,29]. IBC-related overdiagnosis estimates in the same population and definition decreased by 3.0% and 22.7 per 100,000 screened women from 2 to 25 years follow-up [12]. For DCIS overdiagnosis, the overestimation caused by short follow-up was larger in proportion (7%) but smaller in rate (10/100,000), due to the relatively smaller number of DCIS but larger percentage of overdiagnosed cases compared to IBC. This shows the important role of DCIS in breast cancer overdiagnosis compared to IBC, as a relatively large proportion of diagnosed DCIS is overdiagnosed, where the risk for a women participating

in screening to be overdiagnosed with DCIS is relatively low. For IBC-related overdiagnosis, estimates stabilized at 10 years follow-up [12], where for DCIS a minimum of 20 years follow-up is needed. The longer follow-up for DCIS can be explained by the long period DCIS can remain dormant after which it could progress to IBC or regress [3,17]. Short follow-up (<20 years) has resulted in variation and overestimation of overdiagnosis estimates, and is similar for screening start age and DCIS grade.

Age and grade influence overdiagnosis estimates, with the proportion overdiagnosed DCIS over seven times higher for women starting screening at age 74 compared to age 50 at sufficient follow-up and consistently highest for grade 1 and lowest for grade 3. At follow-up <10 years, the effect of start age on overdiagnosis diminishes. The effect of screen start age on DCIS overdiagnosis was not established in earlier studies. One previous study found lower overdiagnosis for women with DCIS grade 3 aged 50–60 compared to women aged 60–75 [11]. Both estimates show that risk of overdiagnosis increases for older age and decreases for higher DCIS grade. This can be explained by the increased risk of breast cancer at older age and lower average progression rate at lower DCIS grade [15]. A higher incidence means more screen-detected DCIS at older age, while the lower progression rate indicates DCIS has a smaller probability of progression to IBC, resulting in a higher overdiagnosis risk for older age and lower DCIS grade. Therefore, age and grade are important factors to take into account in overdiagnosis estimates, and women should be informed that starting screening at an early age lowers risk of overdiagnosis. In addition, this study showed the effect of age on DCIS overdiagnosis perseveres below 50, and hardly any overdiagnosis presents at age 40–49, in line with previous research [29]. Therefore, balance between benefits and harms of screening seems to improve for lower ages, which is also in line with worldwide guidelines of screening from age 40–75 [27,28].

Screening setting influences overdiagnosis, with shorter screen interval and higher compliance resulting in a higher proportion of overdiagnosed DCIS and a lower overdiagnosis rate. The influence of screen interval on overdiagnosis estimates increased with higher compliance. The increased proportion is in line with a previous study which showed that shorter screen interval increased proportion of total breast cancer overdiagnosis [14]. Furthermore, lower compliance (80% versus 100%) caused a 5% decrease of total breast cancer overdiagnosis [14]. For DCIS only, this same decrease in compliance was associated with a 2% decrease in overdiagnosis. With compliance dropping in multiple countries, <80% in the Netherlands [16], the balance between benefits and harms of screening will be affected. Therefore, optimal screening setting should be evaluated with current compliance and efforts should be made to promote screening participation. Also, there was an opposite effect of screening setting on overdiagnosis rate versus proportion. Both a shorter interval and a higher compliance result in more screening, thus more screen-detected tumours, which include relatively more tumours that may not have been diagnosed without screening, resulting in the increased proportion. However, screening more also means more women who do not have a tumour, causing the decreased overdiagnosis rate. Therefore, it is important to report both overdiagnosis proportion and rate to fully understand the effect of determinants on overdiagnosis estimates.

Improved knowledge on the influence of determinants on DCIS overdiagnosis improves accuracy of previous estimates, helps women to understand the risk of overdiagnosis, and can be used in optimization of breast cancer screening and treatment policy. For accurate overdiagnosis estimates, DCIS grade, age, screening setting, and adequate follow-up time should be included. For example, to inform women on the consequences of starting screening at older age, sufficient follow-up time is even more important, as overdiagnosis is more likely in older women, but this effect is not visible with insufficient follow-up time. In addition, an accurate definition should be chosen to provide estimates [10], and both overdiagnosis proportion and rate should be reported. A good understanding of overdiagnosis and all factors that influence it will allow women and policy makers to make informed decisions.

As DCIS is always treated upon diagnosis, DCIS overdiagnosis results in overtreatment [3,5]. The golden standard of DCIS diagnosis is histopathological confirmation using a biopsy and pathological analysis to confirm a positive mammography screening [30]. In recent literature, artificial intelligence assisted histologic diagnosis has been suggested to assist pathologists to increase diagnostic accuracy and quicken the pathological evaluation [30]. The current standard of care of treatment upon detection has been under discussion [3,5]. Randomized controlled trials have been set up to explore active surveillance as an alternative [3]. If diagnosis or treatment standards change, it will be important to assess both overdiagnosis and overtreatment.

This study has some strengths and limitations. First, this study successfully quantified the influence of determinants on DCIS overdiagnosis but did not consider IBC-related overdiagnosis. IBC was an endpoint in the model and the influence of determinants on IBC-related overdiagnosis was not determined. However, previous literature has focused on IBC-related overdiagnosis. Knowing the influence of determinants on overdiagnosis, the next step is to optimize screening weighing all major benefits and harms, including both DCIS and IBC-related overdiagnosis. In addition, direct progression from healthy to IBC was not included in SimDCIS, as the main goal of the model was to estimate DCIS overdiagnosis. Although DCIS overdiagnosis is not directly affected by the number of IBC without DCIS as a precursor, the population of healthy people is larger without the possibility for direct progression. Therefore, there could be bias in our estimation of DCIS overdiagnosis. Sensitivity analysis showed an underestimation of approximately 1% in Dutch screening setting (S6 Table). Second, the SimDCIS model was extensively validated previously and shown to make reliable estimates in context of screening in both Dutch and UK setting [10,15]. However, comparisons with other programs should be done with care, as overdiagnosis estimates should be made for a specific setting. Results of the Dutch screening setting and its variations will likely be comparable to other well-implemented screening programs but are not suited for areas without screening. The impact of a specific factor on the overdiagnosis estimate is, however, expected to be similar. Also, the estimated follow-up time needed for a reliable DCIS overdiagnosis estimate is not expected to vary for other screening settings. Third, this study focused on the general population and did not consider high-risk women. Although, high-risk women may require earlier and more frequent screening, no significant change in overdiagnosis is expected, as the incidence of DCIS in this subgroup is relatively low. Fourth, this study quantifies the influence of DCIS grade, follow-up time, screen start age, interval, and compliance, but there are more possible determinants of overdiagnosis, such as breast density and screening modality. For this study, only mammography was simulated, and breast density was not yet included in the model. Future research should focus on quantification of the influence of other possible determinants.

Accurate overdiagnosis estimates will enable considering overdiagnosis as harm in optimization of screening. To obtain accurate estimates, an improved understanding of the influence of determinants on overdiagnosis estimates is needed and both DCIS and IBC should be considered. Therefore, in previous studies, we have developed a model for DCIS [15], systematically reviewed existing literature and models on DCIS overdiagnosis [17], assessed the influence of the definition of overdiagnosis and DCIS grade on overdiagnosis estimates [10], and determined adequate follow-up time for IBC [12]. In addition to this, the current study focused on quantification of the influence of follow-up time and screening setting on DCIS overdiagnosis. In future research, important remaining determinants should be quantified, such as breast density and screening modality, and breast cancer screening should be optimized considering both DCIS and IBC-related overdiagnosis.

## Conclusion

In conclusion, reliable estimates of overdiagnosis can only be made with consideration of DCIS grade, age, screening setting, and a minimum of 20 years follow-up. Women and policy makers should be informed on how to lower risk of overdiagnosis, such as lower screen start age and higher compliance. Optimization of screening should be evaluated weighing benefits and harms, including DCIS and IBC-related overdiagnosis, and using all important determinants, and screening from age 40–49 could be explored, as it caused little to no increase in DCIS overdiagnosis. Future results from randomized controlled trials might provide opportunities to further improve overdiagnosis estimates. Meanwhile, future research should focus on quantification of other determinants to enable optimization of breast cancer screening taking into account both DCIS and IBC-related overdiagnosis.

## Supporting information

**S1 Fig. Screen start age and DCIS overdiagnosis with varying follow-up time.** DCIS overdiagnosis rate (per 100,000 women screened) after a single screen at age 50–74 years at 25 years follow-up time in Dutch screening setting (biennial mammography, 76% compliance).
(TIF)

**S1 File. Overdiagnosis estimation: calculation main outcomes.**
(DOCX)

**S1 Table. Follow-up time and DCIS overdiagnosis.**
(DOCX)

**S2 Table. Screen start age and DCIS overdiagnosis: age 40–49.**
(DOCX)

**S3 Table. Screen start age and DCIS overdiagnosed proportion by DCIS grade.**
(DOCX)

**S4 Table. Screen start age and DCIS overdiagnosis rate by DCIS grade.**
(DOCX)

**S5 Table. Screening interval and compliance and DCIS overdiagnosis.**
(DOCX)

**S6 Table. Sensitivity analysis: exclusion of direct progression to IBC.**
(DOCX)

## Author contributions

**Conceptualization:** Keris Poelhekken, Marcel J.W. Greuter, Geertruida H. de Bock.

**Data curation:** Keris Poelhekken, Marcel J.W. Greuter.

**Formal analysis:** Keris Poelhekken, Marcel J.W. Greuter, Monique D. Dorrius, Geertruida H. de Bock.

**Investigation:** Keris Poelhekken, Marcel J.W. Greuter, Monique D. Dorrius, Geertruida H. de Bock.

**Methodology:** Keris Poelhekken, Marcel J.W. Greuter, Monique D. Dorrius, Geertruida H. de Bock.

**Project administration:** Keris Poelhekken, Marcel J.W. Greuter, Geertruida H. de Bock.

**Resources:** Keris Poelhekken, Marcel J.W. Greuter, Monique D. Dorrius, Geertruida H. de Bock.

**Software:** Keris Poelhekken, Marcel J.W. Greuter, Monique D. Dorrius, Geertruida H. de Bock.

**Supervision:** Marcel J.W. Greuter, Monique D. Dorrius, Geertruida H. de Bock.

**Validation:** Keris Poelhekken, Marcel J.W. Greuter, Monique D. Dorrius, Geertruida H. de Bock.

**Visualization:** Keris Poelhekken, Marcel J.W. Greuter, Monique D. Dorrius, Geertruida H. de Bock.

**Writing – original draft:** Keris Poelhekken.

**Writing – review & editing:** Keris Poelhekken, Marcel J.W. Greuter, Monique D. Dorrius, Geertruida H. de Bock.

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
