## [Decision Letter · Decision Letter 0]

30 Sep 2025

Dear Dr. Poelhekken,

Thank you for submitting your manuscript to PLOS ONE. After careful consideration, we feel that it has merit but does not fully meet PLOS ONE’s publication criteria as it currently stands. Therefore, we invite you to submit a revised version of the manuscript that addresses the points raised during the review process.

We look forward to receiving your revised manuscript.

Kind regards,

George Kuryan

Academic Editor

PLOS ONE

Journal Requirements:

2. Please note that PLOS One has specific guidelines on code sharing for submissions in which author-generated code underpins the findings in the manuscript. In these cases, all author-generated code must be made available without restrictions upon publication of the work. Please review our guidelines at https://journals.plos.org/plosone/s/materials-and-software-sharing#loc-sharing-code and ensure that your code is shared in a way that follows best practice and facilitates reproducibility and reuse.

4. Please note that funding information should not appear in any section or other areas of your manuscript. We will only publish funding information present in the Funding Statement section of the online submission form. Please remove any funding-related text from the manuscript.

Reviewers' comments:

Reviewer's Responses to Questions

**Comments to the Author**

1. Is the manuscript technically sound, and do the data support the conclusions?

Reviewer #1: Partly

Reviewer #2: Yes

2. Has the statistical analysis been performed appropriately and rigorously?

Reviewer #1: Yes

Reviewer #2: Yes

3. Have the authors made all data underlying the findings in their manuscript fully available?

Reviewer #1: Yes

Reviewer #2: Yes

4. Is the manuscript presented in an intelligible fashion and written in standard English?

Reviewer #1: Yes

Reviewer #2: Yes

Reviewer #1: The main objective of the paper is to use the SimDCIS248 model for simulations aimed at balancing the benefits and harms of screening and avoiding overdiagnosis. However, the only significant conclusion drawn was that, in order to reduce overdiagnosis of DCIS, at least 20 years of follow-up are required. This does not appear to offer any major improvement to the current screening system, and the usefulness of the evaluation framework is questionable. Furthermore, the authors have already submitted similar findings to another journal, which further reduces the value of publishing this paper on PLOS ONE.

Reviewer #2: I read with interest the article entitled “Influence of follow-up, screening age, interval, and compliance on overdiagnosis of ductal carcinoma in situ (DCIS): a modelling study.” The study addresses an important topic; however, I would like to highlight the following observations and suggestions that require clarification before the manuscript may be considered for publication:

1. Overall, the manuscript is well written and provides an evaluation of factors in breast cancer screening that may contribute to the overdiagnosis of DCIS.

2. The title is concise, accurate, and appropriate for the study.

3. The abstract adequately summarizes the manuscript, covering the background, methodology, results, and conclusions.

4. The introduction is clearly structured and appropriate.

5. In the Materials & Methods section, the screening age and frequency appear to have been chosen arbitrarily and do not correspond with the recommendations of either the American Cancer Society or the European Commission Initiative on Breast Cancer (ECIBC).

6. Biennial screening is not endorsed by these societies, and higher compliance rates are generally not sustainable over long-term follow-up.

7. While patients with a positive family history may require earlier and more frequent screening, the incidence of DCIS in this subgroup is relatively low.

8. The imaging modality used for screening (mammography, ultrasonography, or MRI) has not been specified. As this significantly impacts the likelihood of overdiagnosis, it should be clarified.

9. I am unable to comment on the SimDCIS tool.

10. The discussion is generally appropriate; however, it should also address the interventions and the gold standard histopathological methods required to confirm a diagnosis of DCIS.

11. The conclusion is consistent with the study findings.

12. The references are recent and relevant.

**Do you want your identity to be public for this peer review?** For information about this choice, including consent withdrawal, please see our Privacy Policy

Reviewer #1: No

Reviewer #2: **Yes:** GYAN CHAND

---

## [Author Response · Author response to Decision Letter 1]

29 Oct 2025

Dear reviewers and editor,

We want to thank you for your constructive comments.

All comments were addressed and a point-by-point response can be found in the Response to Reviewers file.

Kind regards,

Keris Poelhekken

---

## [Decision Letter · Decision Letter 1]

1 Dec 2025

Dear Dr. Poelhekken,

Thank you for submitting your manuscript to PLOS ONE. After careful consideration, we feel that it has merit but does not fully meet PLOS ONE’s publication criteria as it currently stands. Therefore, we invite you to submit a revised version of the manuscript that addresses the points raised during the review process.

We look forward to receiving your revised manuscript.

Kind regards,

Kuryan George

Academic Editor

PLOS ONE

Journal Requirements:

Reviewers' comments:

Reviewer's Responses to Questions

**Comments to the Author**

Reviewer #3: (No Response)

2. Is the manuscript technically sound, and do the data support the conclusions?

Reviewer #3: Yes

3. Has the statistical analysis been performed appropriately and rigorously?

Reviewer #3: Yes

4. Have the authors made all data underlying the findings in their manuscript fully available?

Reviewer #3: Yes

5. Is the manuscript presented in an intelligible fashion and written in standard English?

Reviewer #3: Yes

Reviewer #3: There are many definitions of overdiagnosis, the authors need to justify why they chose the one mentioned in the manuscript.

The denominator size has been influenced by excluding direct progression to IBC. The authors need to comment on the magnitude of this bias.

The clarity on generalisability of this study needs to be better.

In table-1 better to avoid arrows to depict the increasing and decreasing trend.

**Do you want your identity to be public for this peer review?** For information about this choice, including consent withdrawal, please see our Privacy Policy

Reviewer #3: **Yes:** Gurushankari Balakrishnan

---

## [Author Response · Author response to Decision Letter 2]

12 Dec 2025

Thank you for taking the time to review this manuscript. Please find the detailed responses in the attached file 'Response to Reviewer' and the corresponding revisions in track changes in the re-submitted files.

---

## [Editor Report · Decision Letter 2]

6 Jan 2026

Influence of follow-up, screening age, interval, and compliance on overdiagnosis of ductal carcinoma in situ (DCIS): a modelling study

PONE-D-25-39133R2

Dear Dr. Poelhekken

We’re pleased to inform you that your manuscript has been judged scientifically suitable for publication and will be formally accepted for publication once it meets all outstanding technical requirements.

Kind regards,

George Kuryan

Academic Editor

PLOS One
---

## [Editor Report · Acceptance letter]

PONE-D-25-39133R2

PLOS One

Dear Dr. Poelhekken,

I'm pleased to inform you that your manuscript has been deemed suitable for publication in PLOS One. Congratulations! Your manuscript is now being handed over to our production team.

Kind regards,

on behalf of

Professor George Kuryan

Academic Editor

PLOS One